# POINTWISE CONVERGENCE IN GAMES WITH CONFLICTING INTEREST

## ABSTRACT

In this work, we introduce the concept of non-negative weighted regret, an extension of non-negative regret Anagnostides et al. (2022) in games. Investigating games with non-negative weighted regret helps us to understand games with conflicting interests, including harmonic games and important classes of zero-sum games. We show that optimistic variants of classical no-regret learning algorithms, namely optimistic mirror descent (OMD) and optimistic follow the regularized leader (OFTRL), converge to an $\epsilon$-approximate Nash equilibrium at a rate of $O(1/\epsilon^2)$. Consequently, they guarantee pointwise convergence to a Nash equilibrium if there are only finitely many Nash equilibria in the game. These algorithms are robust in the sense the convergence holds even if the players deviate from prescribed strategies, as long as the corruption level remains finite. Our theoretical findings are supported by empirical evaluations of OMD and OFTRL on the game of matching pennies and harmonic game instances.

## 1 INTRODUCTION

A central question in the study of strategic learning dynamics is the evolution of strategies through iterative interactions: how do these strategies develop over time, and what conditions determine their convergence to equilibrium versus exhibiting recurrent patterns? The challenge is fundamentally rooted in computational complexity theory, as finding Nash equilibria has been proved to be PPAD-hard Daskalakis et al. (2009)—a complexity classification that suggests the widespread belief in the non-existence of efficient polynomial-time algorithms for equilibrium computation in arbitrary games.

Nevertheless, this computational intractability does not uniformly apply across all game classes. Zero-sum games, for instance, present a notably more optimistic scenario Daskalakis & Panageas (2019); Mertikopoulos et al. (2019); Wei et al. (2020); Daskalakis et al. (2020); Wei et al. (2021); Yang & Ma (2022); Leonardos et al. (2022a); Anagnostides et al. (2022). It is shown that the optimistic gradient descent can find the Nash equilibrium. This is later extended to classic no-regret learning algorithms, the online mirror descent (MD) and the follow the regularized leader (FTRL), by developing optimistic variants of them.

The taxonomy of finite games reveals two principal components: potential games and harmonic games. Potential games, characterized by a potential function that captures aligned player interests, have been extensively studied with numerous convergence results Anagnostides et al. (2022); Cominetti et al. (2010); Palaiopanos et al. (2017); Heliou et al. (2017); Daskalakis et al. (2011); Syrgkanis et al. (2015); Chen & Peng (2020); Hsieh et al. (2021); Dong et al. (2024). In contrast, harmonic games embody the competitive aspects of strategic interaction, where any unilateral strategy adjustment invariably creates incentives for other players to respond with counteracting deviations. While the dynamics of learning algorithms in harmonic games generally yield negative results Letcher et al. (2019); Legacci et al. (2024b;a), certain optimistic variants of regularized learning algorithms have demonstrated success Legacci et al. (2024a). It has been shown that all finite games can be decomposed into a potential and a harmonic component Abdou et al. (2022). Although convergent algorithms are both found for potential and harmonic games, the analysis for them are drastically different.

A promising starting point for unifying the positive results of learning dynamics in potential and harmonic games may lie in examining the dynamics within zero-sum games. Zero-sum games, such

Table 1: Overview of results in games with non-negative regret and harmonic games

|  | Games with non-negative regret | Harmonic games |
|---|---|---|
| Anagnostides et al. (2022) | - $O(1/\sqrt{T})$ to approximate Nash equilibrium 
 - No pointwise convergence | - No Guarantee |
| Legacci et al. (2024a) | - No Guarantee | - Asymptotic convergence to the set of NE |
| Ours | - $O(1/\sqrt{T})$ to approximate Nash equilibrium 
 - Asymptotic convergence to set of NE 
 - Pointwise convergence to NE if the set of NE is discrete, 
 such as games with finitely many NE | |

as the classic example of matching pennies, epitomize conflict in strategic interactions. Notably, if a zero-sum game possesses a fully mixed Nash equilibrium, then it is harmonic Legacci et al. (2024a). However, the relationship between zero-sum and harmonic games is nuanced; not all zero-sum games are harmonic, and some can be fully potential. While the relation between zero-sum and harmonic games is not immediately clear, some variants of the optimistic no-regret learning algorithms are found to be convergent in both of them Daskalakis & Panageas (2019); Legacci et al. (2024a).

In this work, we introduce the concept of non-negative weighted regret, which encompasses harmonic games and important classes of zero-sum games, such as polymatrix zero-sum games. This framework also extends to constant-sum polymatrix games. The notion of weighted regret builds upon the concept of non-negative regret introduced in Anagnostides et al. (2022). We focus on the analysis of optimistic no-regret learning algorithms, specifically optimistic mirror descent (OMD) and optimistic follow the regularized leader (OFTRL), within the context of games characterized by non-negative weighted regret. We show that both algorithms can find an $\epsilon$-approximate Nash equilibrium in $O(1/\epsilon^2)$ iterations. Moreover, we establish pointwise convergence of the algorithms to the set of Nash equilibrium, which is the first result of this kind for harmonic games and games with non-negative regrets. When the set of Nash equilibrium is discrete, we show that the iterates converge to a Nash equilibrium of the game, which recovers the convergence results in two-player zero-sum games Daskalakis & Panageas (2019). Table 1 summarizes our results and comparison with the existing results.

Additionally, we investigate the dynamics of OMD and OFTRL when players are permitted to deviate from their algorithmically determined strategies by a finite cumulative amount. We show that when the cumulative deviation is finite, our algorithm still enjoys convergence in the class of games with non-negative weighted regret. This allows us to lift the assumption that players will adhere to their prescribed algorithms in practical problems.

Finally, to substantiate our theoretical findings, we conduct numerical simulations to evaluate the performance of OMD and OFTRL. These simulations validate the practical efficacy of our algorithms and provide empirical support for our theoretical claims.

## 2 RELATED WORKS

**Learning in games with non-negative regrets and zero-sum games** For a constrained zero-sum game with a unique Nash equilibrium, Daskalakis & Panageas (2019) gives an asymptotic last-iterate convergence for optimistic multiplicative weight update. This result is then improved by Wei et al. (2020) to a linear last-iterate convergence rate. However, their result is problem-dependent on a condition number like quantity. Follow-up works then investigated optimistic learning algorithms with vanishing learning rates Mertikopoulos et al. (2019), and with different variants of zero-sum games such as with a Markovian setting Daskalakis et al. (2020); Wei et al. (2021); Yang & Ma (2022); Leonardos et al. (2022b). Some of these results are then generalized in Anagnostides et al. (2022), which proposed the notion of games with nonnegative regret that includes many classes of zero-sum games.

**Learning in harmonic games** When the actions are not constrained to the probability simplex (differential games), Letcher et al. (2019) showed that the gradient dynamic is insufficient to find any stable point of the Hamiltonian game (the analog of harmonic games in differential games). Leveraging the harmonic/potential decomposition, they proposed a Symplectic Gradient Adjustment (SGA) method to find stable points in differential games. In normal-form games, where the actions are confined to the probability simplex, Legacci et al. (2024b) showed the first negative result of the classic no-regret learning. They studied the dynamic induced by the exponential weight algorithm, and showed that the dynamic can be recurrent in harmonic games. This result is later extended by Legacci et al. (2024a), which showed that the classic followed the regularized leader method (of which the exponential weight algorithm is an instance of), is recurrent in harmonic game. However, they showed that by extrapolating the gradients, which includes the variant of optimistic follow the regularized leader, the induced dynamic can converge to the set of Nash equilibrium asymptotically.

## 3 PRELIMINARIES

**Notation** Throughout the paper, we use $\| \cdot \|$ to denote an ambient norm on $\mathbb{R}^d$ and use $\| \cdot \|_*$ to denote the corresponding dual norm. We use $\Delta(\cdot)$ to denote the probability simplex on a finite set, i.e. $\Delta(A_i) = \{x \in \mathbb{R}^{|A_i|}_{\geq 0} : \sum_{a_i \in A_i} x(a_i) = 1\}$.

In this paper, we consider finite normal-form games (NFGs).

### 3.1 NORMAL-FORM GAME

A normal-form game $\Gamma = (n, A, u)$ consists of $n$ players. Each player $i$ has a set of feasible actions $A_i$, with the joint action space denoted as $A = \prod_{i \in [n]} A_i$. Players can adopt randomized strategies $x_i \in \Delta(A_i)$, where $x_i(a_i)$ represents the probability of selecting action $a_i$. The joint action of all players is denoted as $a = (a_1, \ldots, a_n)$, while the joint randomized strategy is represented as $x = (x_1, \ldots, x_n)$.

Each player has a utility function $u_i : A \to [-1, 1]$, where the range is restricted to $[-1, 1]$ for simplicity. Under a joint randomized strategy $x = (x_1, \ldots, x_n)$ with $x_i \in \Delta(A_i)$, the expected utility for player $i$ is given by $u_i(x) = \mathbb{E}_{a \sim x}[u_i(a)] = \sum_{a_i \in A_i} u_i(a_i, x_{-i})x_i(a_i)$, where $x_{-i}$ denotes the joint strategy of all players except player $i$.

The payoff field for an individual player $i$ is defined as $v_i(x) = (u_i(a_i, x_{-i}))_{a_i \in A_i}$, and the overall game's payoff field is $v(x) = (v_1(x), \ldots, v_n(x))$. With this, the utility $u_i$ can then be expressed as $u_i(x) = \langle v_i(x), x_i \rangle$.

### 3.2 SOLUTION CONCEPTS

A popular solution concept of the normal-form game is Nash equilibrium, which is defined as the following.

**Definition 3.1** (Nash equilibrium). *A strategy $x^* = (x_1^*, \ldots, x_n^*) \in \Delta(A)$ is called a **Nash equilibrium** if, for all players $i \in [n]$, it holds that*

$$u_i(x_i^*, x_{-i}^*) \geq u_i(x_i, x_{-i}^*), \quad \forall x_i \in \Delta(\mathcal{A}_i).$$

Equivalently, the Nash equilibrium can be characterized in terms of the payoff field. Specifically, if $x^*$ is a Nash equilibrium, then

$$\langle v(x^*), x - x^* \rangle \leq 0, \forall x \in \Delta(A).$$

In learning algorithms, agents iteratively approach the Nash equilibrium by progressively refining their approximations. The concept of an approximate Nash equilibrium is defined as follows.

**Definition 3.2** ($\epsilon$-approximate Nash equilibrium). *A strategy $x^* = (x_1^*, \ldots, x_n^*) \in \Delta(A)$ is called an $\epsilon$-**approximate Nash equilibrium** if, for all players $i \in [n]$, it satisfies*

$$u_i(x_i^*, x_{-i}^*) \geq u_i(x_i, x_{-i}^*) - \epsilon, \quad \forall x_i \in \Delta(\mathcal{A}_i).$$

### 3.3 HARMONIC GAMES

The harmonic game is a class of normal-form games designed to model scenarios where players' interests are inherently anti-aligned Candogan et al. (2011). This is in direct contrast to potential games, which capture situations where players' interests are aligned and their collective behavior can be described by a single global potential function Monderer & Shapley (1996). It has been shown in Candogan et al. (2011); Abdou et al. (2022) that every normal-form game can be uniquely decomposed into a direct sum of a potential game and a harmonic game. This decomposition provides a structured way to analyze games by separating the aligned and anti-aligned components of players' utilities. The decomposition is unique up to affine transformations of the utility functions, meaning that any linear scaling or shifting of the utilities does not affect the equilibrium strategies.

Following the definition of harmonic games from previous work

**Definition 3.3** (Abdou et al. (2022); Legacci et al. (2024a)). *A Norm-form game* $\Gamma = (n, A, u)$ *is said to be a harmonic game if there exists a collection of weights* $\mu_{i,a_i} \in (0, \infty)$, $a_i \in A_i$, $i \in [n]$ *such that* $\sum_{i \in [n]} \sum_{b_i \in A_i} \mu_{i,a_i} (u_i(a) - u_i(b_i, a_{-i})) = 0$, *for all of* $a \in A$.

Conceptually, the players' interests in a harmonic game are fundamentally anti-aligned. Specifically, if any player considers deviating toward a particular action, there will always exist other players with an incentive to deviate away from the resulting strategy profile.

## 4 NO-REGRET LEARNING

In classic online learning, an important measure to evaluate the performance of an algorithm is the notion of regret. Regret quantifies the cumulative difference between the utility achieved by the algorithm's chosen actions and the maximum utility that could have been obtained using a fixed action selected in hindsight. Although the original concept of regret was developed in the context of single-player learning, it accommodates scenarios where the utility function varies over time in the most adversarial manner.

In the multi-player learning setting, the utility function remains time-varying from an individual player's perspective, as it is influenced by the evolving strategies of other players. However, regret can be more favorable in this setting compared to single-player scenarios, as the utilities may not vary in the worst case when all players follow the same algorithm.

**Definition 4.1.** *The regret of player* $i$ *is defined as* $\text{Reg}_i^T = \max_{x_i^* \in \Delta(A_i)} \left\{ \sum_{t=1}^T \langle x_i^*, v_i^t \rangle \right\} - \sum_{t=1}^T \langle x_i^t, v_i^t \rangle$, *where* $v_i^t = v_i(x^t)$ *is the payoff vector to player* $i$ *when players pay the sequence of strategy* $x^t$.

An important class of algorithms that achieves a sum of $O(1)$ regret between players is the optimistic online learning algorithms Syrgkanis et al. (2015). This includes the Optimistic Mirror Descent (OMD) and the Optimistic Follow the Regularized Leader (OFTRL) algorithms. Unlike the classic Online Mirror Descent or Online Follow the Regularized Leader algorithms, the optimistic variants take advantage of the fact that utilities may not vary in the worst possible way, allowing them to learn the predictable sequence of changing utilities. As a result, these variants enjoy a regret bound that is influenced by the variation in utility (RVU), a property that has been identified as crucial for achieving small regret in game-theoretic settings Rakhlin & Sridharan (2013); Syrgkanis et al. (2015).

**OMD** Formally, the optimistic mirror descent takes the following steps to select the strategies. We fix a player $i$ for the update steps for clearer presentation. Let $D_{R_i}$ denote the Bregman divergence with regularizer $R_i$, which we assume to be 1-strongly convex with respect to $\|\cdot\|$. Define $x_i^0 = g_i^0 = \text{argmin}_{x_i \in \Delta(A_i)} R_i(x_i)$.

$$x_i^{t+1} = \underset{x_i \in \Delta(A_i)}{\text{argmax}} \, \eta_i^{t+1} \langle x_i, v_i^t \rangle - D_{R_i}(x_i, g_i^t), \tag{1}$$

$$v_i^{t+1} = v_i(x^{t+1}), \quad g_i^{t+1} = \underset{g_i \in \Delta(A_i)}{\text{argmax}} \, \eta_i^{t+1} \langle g_i, v_i^{t+1} \rangle - D_{R_i}(g_i, g_i^t).$$

**OFTRL**   Define $\hat{x}_i^0 = \operatorname{argmin}_{x_i \in \Delta(A_i)} R_i(x_i)$.

$$\hat{x}_i^{t+1} = \operatorname*{argmax}_{x_i \in \Delta(A_i)} \eta_i \left\langle x_i, \hat{v}_i^t + \sum_{s=1}^t \hat{v}_i^s \right\rangle - R_i(x_i), \quad \hat{v}_i^{t+1} = v_i(\hat{x}^{t+1}). \quad (2)$$

Beyond achieving $O(1)$ total regret, these optimistic algorithms have also been shown to converge quickly to the optimal welfare in smooth games Syrgkanis et al. (2015). When the total regret of the game is non-negative, i.e., $\sum_{i \in [n]} \operatorname{Reg}_i^T \geq 0$, these algorithms can also find a $\epsilon$-approximate Nash equilibrium after a sufficient number of iterations Anagnostides et al. (2022).

# 5   GAMES WITH NON-NEGATIVE WEIGHTED REGRETS

It has been demonstrated that the class of games with non-negative regrets encompasses important variants of zero-sum games, including two-player zero-sum games and polymatrix zero-sum games. Conceptually, these zero-sum games share similarities with harmonic games, as both involve naturally conflicting interests between players. However, the standard notion of non-negative regret does not fully capture the class of harmonic games. To address this gap, we introduce the concept of weighted regret, which provides a bridge between zero-sum games and harmonic games.

We further defined the weighted regret of all players as

**Definition 5.1.** *The weighted regret is defined as* $\operatorname{mReg} = \sum_{i=1}^n \operatorname{mReg}_i^T$, *where* $\operatorname{mReg}_i^T = \max_{x^* \in \Delta(A_i)} \sum_{t=1}^T m_i \langle x^* - x_i^t, v_i^t \rangle$, *and* $m_i$ *is a non-zero weight and* $v_i^t = v_i(x^t)$.

A game is with non-negative weighted regret if the weighted sum over all players' regret is non-negative.

**Definition 5.2.** *A game has non-negative weight regret, if* $\exists m \in \mathcal{R}_{++}^n$,

$$\sum_{i=1}^n \operatorname{mReg}_i^T \geq 0, \quad \forall \{x^t\}_{t=1}^T \in \Delta(A), T \geq 1.$$

It is obvious that if a game has non-negative regret, then it has non-negative weighted regret (just let $m = (1, \cdots, 1)$). However, the reverse is not true. Figure 1 specifies a harmonic game instance, whose total regret is negative and by Lemma 5.1 its weighted regret is non-negative. So our framework naturally includes a broader class of games. Having established the definition of non-negative weighted regret, we can now explore its implications for specific classes of games.

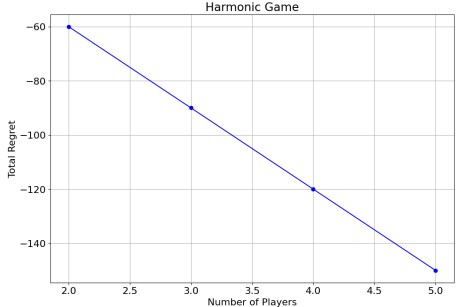

Figure 1: Total regret plot in a Harmonic game where the utility is the collective sum of action minus the individual action. Each point represents the total regret after 10 rounds.

**Lemma 5.1.** *Harmonic games have non-negative weighted regrets.*

Since weighted regret is a natural extension of the concept of non-negative regret, the class of games that exhibit non-negative regret will also inherently have non-negative weighted regret. This provides a broader framework for analyzing a variety of game types within this class.

**Lemma 5.2** (Extension of Proposition 3.2 of Anagnostides et al. (2022)). *The following games also have non-negative weighted regret. 1) Two-player zero-sum games; 2) Polymatrix zero-sum games; 3) Constant-sum Polymatrix games; 4) Strategically zero-sum games; 5) Polymatrix strategically zero-sum games;*

*Proof.* Take $m_i = 1$ and the rest follow from Proposition 3.2 of Anagnostides et al. (2022). □

It is worth noticing that the framework of harmonic game includes some classes two-player zero-sum games. Specifically, a two-player zero-sum game with an interior (fully randomized) Nash

equilibrium $x^*$ is harmonic with the weights $\mu = x^*$. However, there are zero-sum games that are also potential, and have strict (not randomized) equilibrium, which are clearly not captured by the framework of harmonic game.

# 6 Convergence in Games with Non-negative Weighted Regret

Having established that harmonic games, along with other classes of games, enjoy nonnegative weighted regret, we now study the performance of optimistic learning algorithms in these games.

We first focus on optimistic mirror descent, the following theorem formalizes the conditions under which OMD guarantees convergence to an $\epsilon$-approximate Nash equilibrium in games with non-negative weight regret.

**Theorem 6.1.** *If each player employs OMD with*

- *a pair of norms such that $\|x\| \geq c\|x\|_1$, $\|x\|_* \leq c_*\|x\|_\infty$ for some constant $c, c_*$ and for any $x$,*

- *$G_i$-smooth regularizer $R_i$,*

- *non-increasing learning rate with $\eta^1 \leq \frac{c}{4c_*(n-1)}\sqrt{\frac{\underline{m}}{\bar{m}}}$, where $\eta^1 = \max_i \eta_i^1$, $\underline{m} = \min m_i$, $\bar{m} = \max m_i$ and $\eta_i^t \geq \eta_i > 0$.*

*Then if the game has non-negative weight regret, for any $\epsilon > 0$, after $T > \frac{1}{\epsilon^2}\sum_{i=1}^n \frac{8\bar{R}_i m_i \eta^1}{\eta_i \underline{m}}$ iterations, there exists an iterate $x^t$ that is an $\epsilon \cdot \left(c_* + 2\max_{i\in[n]}\left\{\frac{G_i \cdot \Omega_i}{\eta_i}\right\}\right)$-approximate Nash Equilibrium, where $\Omega_i = \sup_{x,y\in\Delta(A_i)}\|x - y\|$.*

We note that the above theorem can be achieved with any initialization of the OMD algorithm. We define the initial point of OMD in Equation 1 to facilitate the later analysis of equivalence between OMD and FTRL. Beyond the finite iterate guarantee, we show that the iterate converges to the set of Nash equilibria.

**Theorem 6.2.** *Suppose $x^t$ is the sequence of strategies output by OMD and the conditions specified in Theorem 6.1 are satisfied, then $x^t$ converges to the set of Nash equilibrium of the game.*

We remark that our analysis is drastically different and more general from Legacci et al. (2024a), which allows our results to be applicable to a broader class of games with non-negative weighted regret, while theirs are only applicable to harmonic games. Our results also give a stronger theoretical guarantee when the distance between the equilibrium is non-zero, in which case the iterates of the OMD converge to a Nash equilibrium.

**Definition 6.1.** *We say the set of Nash equilibrium of the game $E$ is discrete if $d > 0$, where d is defined as $d = \inf_{x,y\in E, x\neq y}\|x - y\| > 0$, if $E$ at least has two points or $d = 1$, if $E$ only has one point.*

**Theorem 6.3.** *Suppose $x^t$ is the sequence of strategies generated by OMD and the conditions specified in Theorem 6.1 are satisfied. If the set of Nash equilibrium of the game is discrete, then $x^t$ converges to a Nash equilibrium of the game.*

Compared to the convergence results for games with non-negative regrets, Theorem 6.3 implies pointwise convergence. It is noted in Anagnostides et al. (2022) (Remark A.15) that their technique cannot imply pointwise convergence. Even in the two-player zero-sum game, the pointwise convergence result often requires the assumption of unique Nash equilibrium Daskalakis & Panageas (2019) or condition number like quantity for the Nash equilibrium set Wei et al. (2020). Under the unique Nash equilibrium and with Theorem 6.1, the pointwise convergence to the Nash equilibrium is apparent. This is because any convergent subsequence of $x^t$ must converge to the unique Nash equilibrium, and the convergence of $x^t$ follows as $x^t$ is bounded (If a bounded sequence's all convergent subsequences converge to the same point, then this sequence converge to this point). However, when the Nash equilibria are not unique, the subsequence of the $x^t$ can converge to different points, and hence $x^t$ may diverge. To tackle this, we first construct a family of sufficiently small open

balls to cover the Nash equilibrium set by the method of functional analysis. Then we meticulously examined whether $x^t$ (for sufficiently large t) belonged to the same open ball to demonstrate the convergence of $x^t$.

Lastly, we would like to note that compared to the previous convergence results for harmonic games Legacci et al. (2024a), our results further strengthen the convergence to a Nash equilibrium (instead of just converging to the Nash equilibrium set) when $d > 0$. The two analyses are in very different technical routes.

Similarly, we have the following guarantee for OFTRL.

**Theorem 6.4.** *If each player employs OFTRL with*

- *a pair of norms such that $\|x\| \geq c\|x\|_1$, $\|x\|_* \leq c_*\|x\|_\infty$ for some constant $c, c_*$ and for any $x$,*

- *a $G_i$ smooth regularizer $R_i$, and $R_i$ is Legendre with domain $D_i \subseteq \Delta(A_i)$*

- *learning rate $\eta \leq \frac{c}{4c_*(n-1)}\sqrt{\frac{\underline{m}}{\bar{m}}}$, where $\eta = \max_i \eta_i$, $\underline{m} = \min m_i$, $\bar{m} = \max m_i$.*

*Then if the game has non-negative weight regret, for any $\epsilon > 0$, after $T > \frac{1}{\epsilon^2}\sum_{i=1}^{n}\frac{8\bar{R}_i m_i \eta}{\eta_i \underline{m}}$ iterations, there exists an iterate $\hat{x}^t$ that is an $\epsilon \cdot \left(c_* + 2\max_i\left\{\frac{G_i \cdot \Omega_i}{\eta_i}\right\}\right)$-approximate Nash Equilibrium, where $\Omega_i = \sup_{x,y \in \Delta(A_i)} \|x - y\|$.*

Similar to the OMD, when the distance between the equilibrium is non-zero, then the iterates converge to a Nash equilibrium.

**Theorem 6.5.** *Suppose $\hat{x}^t$ is the sequence of strategies output by OFTRL and the conditions specified in Theorem 6.1 are satisfied. If the set of Nash equilibrium of the game is discrete, then $\hat{x}^t$ converges to a Nash equilibrium of the game.*

# 7 LEARNING WITH CORRUPTIONS

One crucial assumption underlying optimistic learning algorithms is that all players adhere to the prescribed strategy, following the algorithm's recommendations. When players deviate from the algorithm's prescribed actions, the algorithm can no longer guarantee convergence to the equilibrium. Such deviations are not uncommon, as individual players may face external constraints or have strategic incentives that lead them to act outside the prescribed strategy.

In scenarios where players deviate from the algorithm, we refer to the learning dynamics as being "corrupted." The extent of this corruption can vary depending on the degree to which players stray from the prescribed actions. The following definition (proposed by Tsuchiya et al. (2024)) provides a formal way to quantify the level of corruption present in the dynamic, allowing us to assess its impact on the convergence properties of the algorithm.

**Definition 7.1.** *A game is said to be a corrupted regime with corruption level $\{C_i\}_{i \in [n]}$ if the strategies committed by the player, $\tilde{x} = \{\tilde{x}_1, \ldots, \tilde{x}_n\}$ deviates from the algorithm output $x = \{x_1, \ldots, x_n\}$ by at most $C_i$, i.e. $\sum_{i=1}^{\infty}\|\tilde{x}_i - x_i\|_1 = C_i < \infty$, for all $i \in [n]$.*

In the corruption setting and under OMD or OFTRL, the strategies played by the players can be denoted as follows. Under OMD, we define define $x_i^0 = g_i^0 = \operatorname{argmin}_{x_i \in \Delta(A_i)} R_i(x_i)$, then

$$\tilde{x}_i^{t+1} = \underset{x_i \in \Delta(A_i)}{\operatorname{argmax}} \eta_i\langle x_i, v_i^t\rangle - D_{R_i}(x_i, g_i^t), \quad x_i^{t+1} = \tilde{x}_i^{t+1} + c_i^{t+1},$$

$$v_i^{t+1} = v_i(x^{t+1}), \qquad g_i^{t+1} = \underset{g_i \in \Delta(A_i)}{\operatorname{argmax}} \eta_i\langle g_i, v_i^{t+1}\rangle - D_{R_i}(g_i, g_i^t).$$

Under OFTRL, we define $y_i^0 = \operatorname{argmin}_{y_i \in \Delta(A_i)} R_i(y_i)$, and $\tilde{y}_i^{t+1} = \operatorname{argmax}_{y_i} \eta_i\left\langle y_i, \hat{v}_i^t + \sum_{s=1}^{t}\hat{v}_i^s\right\rangle - R_i(x_i)$ $y_i^{t+1} = \tilde{y}_i^{t+1} + c_i^{t+1}$, $\hat{v}_i^{t+1} = v_i(y^{t+1})$.

When the corruption level remains finite, we expect that the algorithm can still retain its effectiveness and guarantee convergence to an equilibrium, albeit with some modifications to the convergence rate and the quality of the equilibrium.

**Theorem 7.1.** *If each player employs OMD under corruption with*

- *a pair of norms such that $\|x\| \geq c\|x\|_1$, $\|x\|_* \leq c_*\|x\|_\infty$ for some constant $c, c_*$ and for any $x$,*

- *$G_i$-smooth regularizer $R_i$,*

- *non-increasing learning rate with $\eta^1 \leq \frac{c}{4(n-1)c_*} \cdot \sqrt{\frac{\underline{m}}{3\bar{m}}}$ , where $\eta^1 = \max_i \eta_i^1$ , $\underline{m} = \min m_i$ , $\bar{m} = \max m_i$ and $\eta_i^t \geq \eta_i > 0$.*

*Then if the game has non-negative weight regret, for any $\epsilon > 0$, after $T > \frac{1}{\epsilon^2}\left\{\sum_{i=1}^n \frac{8m_i \cdot \bar{R}_i \eta^1}{\eta_i \cdot \underline{m}} 48(\eta^1)^2 \cdot \frac{\bar{m}}{\underline{m}} c_*^2 (n-1)^2 \sum_{i=1}^n M_i \cdot C_i 8\eta^1 \cdot \frac{\bar{m}}{\underline{m}} \sum_{i=1}^n C_i\right\}$ iterations, there exists an iterate $x^t$ that is an $\max_{i \in [n]} \epsilon \cdot \left(c_* + 2\left\{\frac{G_i \cdot \Omega_i}{\eta_i}\right\}\right) + \|c_i^t\|_1$-approximate Nash Equilibrium, where $\Omega_i = \sup_{x,y \in \Delta(A_i)} \|x - y\|$ and $M_i = \sup_{t \geq 1} c_i^t$.*

When the corruption is finite, we can expect the iterates of OMD to converge to the set of Nash equilibria. The following Theorem gives a guarantee that is similar to that of the non-corrupted case.

**Theorem 7.2.** *Suppose $x^t$ is the sequence of strategies played with OMD under corruptions and the conditions specified in Theorem 7.1 are satisfied, then $x^t$ converges to the set of Nash equilibria of the game.*

Further, when the distances between the Nash equilibria are larger than zero and the corruptions are finite, the iterates of OMD converge to a Nash equilibrium.

**Theorem 7.3.** *Suppose $x^t$ is the sequence of strategies played with OMD under corruptions and the conditions specified in Theorem 7.1 are satisfied. If the set of Nash equilibria of the game is discrete, then $x^t$ converges to a Nash equilibrium of the game.*

To the best of our knowledge, the only previous work on games with corrupted dynamics is Tsuchiya et al. (2024), which proposed a variant of OFTRL that enjoys $O(1)$ regret when the corruption level is small. This implies that their method also converges to a correlated equilibrium at the rate of $O(1/T)$ when the corruption level is small. In comparison, our method gives the first convergence guarantee to a Nash equilibrium under a finite corruption level, which is a much stricter equilibrium than a correlated equilibrium. Our convergence rate is $O(1/\epsilon^2)$ for an $\epsilon$-approximate Nash equilibrium, and our technique implies pointwise convergence.

the previous result was only achieved under OFTRL Tsuchiya et al. (2024). In the non-corrupted case, OFTRL and OMD are equivalent in the sense that they give similar guarantees. The following two Theorem show that this equivalency extends to the corruption case when the corruption level is finite.

**Theorem 7.4.** *If each player employs OFTRL with corruption with*

- *a pair of norms such that $\|y\| \geq c\|y\|_1$, $\|y\|_* \leq c_*\|y\|_\infty$ for some constant $c, c_*$ and for any $y$,*

- *a $G_i$ smooth regularizer $R_i$, and $R_i$ is Legendre with domain $D_i \subseteq \Delta(A_i)$*

- *learning rate $\eta \leq \frac{c}{4c_*(n-1)}\sqrt{\frac{\underline{m}}{3\bar{m}}}$ , where $\eta = \max_i \eta_i$ , $\underline{m} = \min m_i$ , $\bar{m} = \max m_i$.*

*Then if the game has non-negative weight regret, for any $\epsilon > 0$, after $T > \frac{1}{\epsilon^2}\left\{\sum_{i=1}^n \frac{8m_i \cdot \bar{R}_i \eta^1}{\eta_i \cdot \underline{m}} + 48(\eta^1)^2 \cdot \frac{\bar{m}}{\underline{m}} c_*^2 (n-1)^2 \sum_{i=1}^n M_i \cdot C_i + 8\eta^1 \cdot \frac{\bar{m}}{\underline{m}} \sum_{i=1}^n C_i\right\}$ iterations, there exists an iterate $y^t$ that is an $\max_{i \in [n]} \epsilon \cdot \left(c_* + 2\left\{\frac{G_i \cdot \Omega_i}{\eta_i}\right\}\right) + \|c_i^t\|_1$-approximate Nash Equilibrium, where $\Omega_i = \sup_{x,y \in \Delta(A_i)} \|x - y\|$ and $M_i = \sup_{t \geq 1} c_i^t$.*

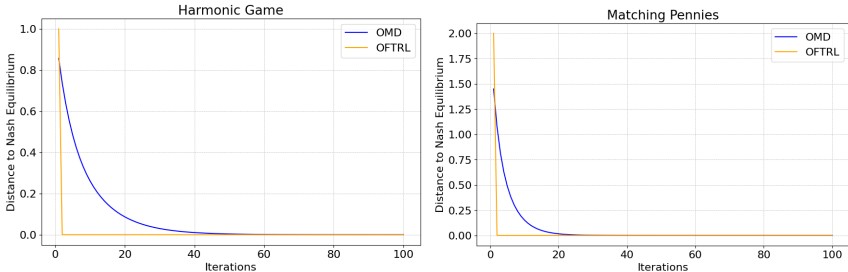

Figure 2: The two plots illustrate the convergence of OMD and OFTRL algorithms towards the Nash equilibrium in the Matching Pennies and the Harmonic game over 100 iterations.

**Theorem 7.5.** *Suppose $y^t$ is the sequence of strategies player with OFTRL under corruptions and the conditions specified in Theorem 7.4 are satisfied. If the set of Nash equilibrium of the game is discrete, then $y^t$ converges to the a Nash equilibrium of the game.*

## 8 EXPERIMENTS

We complement our theoretical findings by empirically evaluating the optimistic algorithms on the Matching pennies, a classic zero-sum game, and a harmonic game. The matching pennies is a two-player zero-sum game where the utility is given by $\begin{bmatrix} -1 & 1 \\ 1 & -1 \end{bmatrix}$. The two-player harmonic game has the utility of $\begin{bmatrix} 1 & 2 \\ 2 & 1 \end{bmatrix}$. In both experiments, the learning rate of OMD and OFTRL are set to $0.1$.

## 9 CONCLUSION AND FUTURE DIRECTIONS

In this work, we propose the notion of non-negative weighted regret, which serves as a framework to encapsulate the harmonic games and important classes of zero-sum games. This notion is an extension of the games with non-negative regret Anagnostides et al. (2022) and helps to further our understanding of the interplay between harmonic games and zero-sum games, which are both games with conflicting interest, but do not have an inclusion relationship. We then study the optimistic variants of the classic no-regret learning algorithms, namely the optimistic mirror descent (OMD) and the optimistic follow the regularized leader (OFTRL) algorithms. We show that both algorithms can converge to $\epsilon$-approximate Nash equilibrium efficiently at a rate of $1/\epsilon^2$. Moreover, our result implies pointwise convergence of a Nash equilibrium when the set of Nash equilibrium is discrete. To our best knowledge, this is the first pointwise convergence result in harmonic games and games with non-negative regret. This convergence holds even if the players do not comply with their prescribed algorithm up to a finite corruption level, which corroborates a wider set of applications.

It is known that the zero-sum game can be potential. Yet the class of potential games does not seem to fit in the framework of non-negative (weighted) regret and it is unclear whether the optimistic algorithms are effective in potential games. An important direction would be to explore the relationship between non-negative (weighted) regret and potential games. While the non-negative (weighted) regret can be used to summarize the convergent results in zero-sum games and harmonic games, it remains in question whether it can be used to summarize the negative behaviors (the recurrent and chaotic behaviors) of algorithms in the two classes of games.

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

## A HARMONIC GAMES

**Lemma 5.1.** *Harmonic games have non-negative weighted regrets.*

*Proof.* From the definition of a harmonic game:

$$\sum_{i=1}^{n} \sum_{b_i \in A_i} \mu_{i,b_i} \left( u_i(a_i, a_{-i}) - u_i(b_i, a_{-i}) \right) = 0, \forall a \in A \,,$$

Multiply both sides by $x_a, \forall x \in \Delta(A)$:

$$x_a \cdot \sum_{i=1}^{n} \sum_{b_i \in A_i} \mu_{i,b_i} \left( u_i(a_i, a_{-i}) - u_i(b_i, a_{-i}) \right) = 0, \forall a \in A \,,$$

Summing it for all $a \in A$, we have that,

$$\sum_{a \in A} x_a \cdot \sum_{i=1}^{n} \sum_{b_i \in A_i} \mu_{i,b_i} \left( u_i(a_i, a_{-i}) - u_i(b_i, a_{-i}) \right) = 0, \forall x \in \Delta(A) \,,$$

Exchange summation sequence, we have that,

$$\sum_{i=1}^{n} \sum_{b_i \in A_i} \mu_{i,b_i} \sum_{a \in A} x_a \cdot \left( u_i(a_i, a_{-i}) - u_i(b_i, a_{-i}) \right) = 0, \forall x \in \Delta(A) \,,$$

And then we have,

$$\sum_{i=1}^{n} \sum_{b_i \in A_i} \mu_{i,b_i} \left( u_i(x_i, x_{-i}) - \sum_{a_i \in A_i} x_{i.a_i} \cdot u_i(b_i, x_{-i}) \right) = 0, \forall x \in \Delta(A) \,, \tag{3}$$

Since $\sum_{a_i \in A_i} x_{i,a_i} = 1$,

$$\sum_{i=1}^{n} \sum_{b_i \in A_i} \mu_{i,b_i} \left( \langle v_i(x), x_i \rangle - u_i(b_i, x_{-i}) \right) = 0, \forall x \in \Delta(A) \,,$$

Let $m_i = \sum_{b_i \in A_i} \mu_{i,b_i} > 0$, then,

$$\sum_{i=1}^{n} m_i \langle v_i(x), x_i \rangle = \sum_{i=1}^{n} \langle v_i(x), \mu_i \rangle, \forall x \in \Delta(A) \,,$$

let $x^* = \left( \frac{\mu_i}{m_i}, \cdots, \frac{\mu_n}{m_n} \right)$, since $x^*_{i,a_i} = \frac{\mu_{i,a_i}}{m_i} \in (0,1)$, and $\sum_{a_i \in A_i} x^*_{i,a_i} = \sum_{a_i \in A_i} \frac{\mu_{i,a_i}}{m_i} = 1$. Therefore, $x^* \in \Delta(A)$. After that, we finally have:

$$
\begin{aligned}
\sum_{i=1}^{n} \mathrm{mReg}_i^T &\geq \sum_{i=1}^{n} \sum_{t=1}^{T} m_i \langle x_i^* - x_i^t, v_i^t \rangle \\
&= \sum_{t=1}^{T} \sum_{i=1}^{n} m_i \langle x_i^*, v_i^t \rangle - \sum_{t=1}^{T} \sum_{i=1}^{n} m_i \langle x_i^t, v_i^t \rangle \\
&= \sum_{t=1}^{T} \sum_{i=1}^{n} \langle \mu_i, v_i^t \rangle - \sum_{t=1}^{T} \sum_{i=1}^{n} \langle \mu_i, v_i^t \rangle \\
&= 0
\end{aligned}
$$

□

## B    ANALYSIS FOR OMD

Define $\bar{R}_i = \max_{x,y \in \Delta(A_i)} D_{R_i}(x, y)$.

**Lemma B.1.**
$$\text{Reg}_i^T \leq \frac{\bar{R}_i}{\eta_i^T} + \sum_{t=1}^{T} \|g_i^t - x_i^t\|\|v_i^t - v_i^{t-1}\|_* - \frac{1}{2} \sum_{t=1}^{T} \frac{1}{\eta_i^t} \left( \|g_i^t - x_i^t\|^2 + \|x_i^t - g_i^{t-1}\|^2 \right) .$$

*Proof.* For $x_i^* \in \Delta(A_i)$,
$$\langle x_i^* - x_i^t, v_i^t \rangle = \langle g_i^t - x_i^t, v_i^t - v_i^{t-1} \rangle + \langle g_i^t - x_i^t, v_i^{t-1} \rangle + \langle x_i^* - g_i^t, v_i^t \rangle ,$$
and by Cauchy-Schwarz inequality
$$\langle g_i^t - x_i^t, v_i^t - v_i^{t-1} \rangle \leq \|g_i^t - x_i^t\|\|v_i^t - v_i^{t-1}\|_* .$$

Let $a^* = \text{argmax}_{a \in A} \eta \langle a, x \rangle - D_R(a, c)$. Then for any $d \in A$, $\langle \eta x - \nabla R(a^*) + \nabla R(c), d - a^* \rangle \leq 0$. Rearranging this, we have
$$\langle d - a^*, x \rangle \leq \frac{1}{\eta} \left[ D_R(d, c) - D_R(d, a^*) - D_R(a^*, c) \right] .$$

Applying this we get
$$\langle g_i^t - x_i^t, v_i^{t-1} \rangle \leq \frac{1}{\eta_i^t} \left[ D_{R_i}(g_i^t, g_i^{t-1}) - D_{R_i}(g_i^t, x_i^t) - D_{R_i}(x_i^t, g_i^{t-1}) \right] ,$$
and
$$\langle x_i^* - g_i^t, v_i^t \rangle \leq \frac{1}{\eta_i^t} \left[ D_{R_i}(x_i^*, g_i^{t-1}) - D_{R_i}(x_i^*, g_i^t) - D_{R_i}(g_i^t, g_i^{t-1}) \right] .$$

Combining these, we have
$$\langle x_i^* - x_i^t, v_i^t \rangle \leq \|g_i^t - x_i^t\|\|v_i^t - v_i^{t-1}\|_* + \frac{1}{\eta_i^t} \left( D_{R_i}(x_i^*, g_i^{t-1}) - D_{R_i}(x_i^*, g_i^t) - \frac{1}{2}\|g_i^t - x_i^t\|^2 - \frac{1}{2}\|x_i^t - g_i^{t-1}\|^2 \right) .$$

And we used the strong convexity of $R_i : D_{R_i}(x, g) \geq \frac{1}{2}\|x - g\|^2$ , for $\forall x, g \in \Delta(A_i)$. Summing over $T$, we have
$$\sum_{t=1}^{T} \langle x_i^* - x_i^t, v_i^t \rangle \leq \frac{1}{\eta_i^1} D_{R_i}(x_i^*, g_i^0) + \sum_{t=2}^{T} \left( \frac{1}{\eta_i^t} - \frac{1}{\eta_i^{t-1}} \right) D_{R_i}(x_i^*, g_i^{t-1}) + \sum_{t=1}^{T} \|g_i^t - x_i^t\|\|v_i^t - v_i^{t-1}\|_*$$
$$- \frac{1}{2} \sum_{t=1}^{T} \frac{1}{\eta_i^t} \left( \|g_i^t - x_i^t\|^2 + \|x_i^t - g_i^{t-1}\|^2 \right)$$
$$\leq \frac{\bar{R}_i}{\eta_i^T} + \sum_{t=1}^{T} \|g_i^t - x_i^t\|\|v_i^t - v_i^{t-1}\|_* - \frac{1}{2} \sum_{t=1}^{T} \frac{1}{\eta_i^t} \left( \|g_i^t - x_i^t\|^2 + \|x_i^t - g_i^{t-1}\|^2 \right) .$$

$\square$

**Lemma B.2.**
$$\text{Reg}_i^T \leq \frac{\bar{R}_i}{\eta_i^T} + \sum_{t=1}^{T} \eta_i^t \|v_i^t - v_i^{t-1}\|_*^2 - \frac{1}{4} \sum_{t=1}^{T} \frac{1}{\eta_i^t} \left( \|g_i^t - x_i^t\|^2 + \|x_i^t - g_i^{t-1}\|^2 \right) .$$

*Proof.* For any $\rho^t > 0$, we have
$$\|g_i^t - x_i^t\|\|v_i^t - v_i^{t-1}\|_* \leq \frac{\rho^t}{2}\|v_i^t - v_i^{t-1}\|_*^2 + \frac{1}{2\rho^t}\|g_i^t - x_i^t\|^2 .$$

Using $\rho^t = 2\eta_i^t$ with Lemma B.1, we have
$$\sum_{t=1}^{T} \langle x_i^* - x_i^t, v_i^t \rangle \leq \frac{\bar{R}_i}{\eta_i^T} + \sum_{t=1}^{T} \eta_i^t \|v_i^t - v_i^{t-1}\|_*^2 - \frac{1}{4} \sum_{t=1}^{T} \frac{1}{\eta_i^t} \left( \|g_i^t - x_i^t\|^2 + \|x_i^t - g_i^{t-1}\|^2 \right) .$$

$\square$

**Lemma B.3.**

$$\mathrm{mReg}_i^T \leq \frac{m_i \bar{R}_i}{\eta_i^T} + m_i \sum_{t=1}^{T} \eta_i^t \|v_i^t - v_i^{t-1}\|_*^2 - \frac{m_i}{4} \sum_{t=1}^{T} \frac{1}{\eta_i^t} \left( \|g_i^t - x_i^t\|^2 + \|x_i^t - g_i^{t-1}\|^2 \right).$$

*Proof.* By the definition of $\mathrm{mReg}_i^T$, we have

$$\mathrm{mReg}_i^T = \max_{x^* \in \Delta(A_i)} \sum_{t=1}^{T} m_i \langle x^*, v_i^t \rangle - \sum_{t=1}^{T} m_i \langle x_i^t, v_i^t \rangle$$

$$\leq \frac{m_i \bar{R}_i}{\eta_i^T} + \sum_{t=1}^{T} \eta_i^t m_i \|v_i^t - v_i^{t-1}\|_*^2 - \frac{m_i}{4} \sum_{t=1}^{T} \frac{1}{\eta_i^t} \left( \|g_i^t - x_i^t\|^2 + \|x_i^t - g_i^{t-1}\|^2 \right),$$

by Lemma B.2. $\qquad\square$

**Lemma B.4** (Claim A.1 of Anagnostides et al. (2022)).

$$\|v_i^t - v_i^{t-1}\|_\infty \leq \sum_{j \neq i} \|x_j^t - x_j^{t-1}\|_1.$$

**Theorem 6.1.** *If each player employs OMD with*

- *a pair of norms such that $\|x\| \geq c\|x\|_1$, $\|x\|_* \leq c_*\|x\|_\infty$ for some constant $c, c_*$ and for any $x$,*

- *$G_i$-smooth regularizer $R_i$,*

- *non-increasing learning rate with $\eta^1 \leq \frac{c}{4c_*(n-1)}\sqrt{\frac{\underline{m}}{\bar{m}}}$, where $\eta^1 = \max_i \eta_i^1$, $\underline{m} = \min m_i$, $\bar{m} = \max m_i$ and $\eta_i^t \geq \eta_i > 0$.*

*Then if the game has non-negative weight regret, for any $\epsilon > 0$, after $T > \frac{1}{\epsilon^2} \sum_{i=1}^{n} \frac{8\bar{R}_i m_i \eta^1}{\eta_i \underline{m}}$ iterations, there exists an iterate $x^t$ that is an $\epsilon \cdot \left( c_* + 2\max_{i \in [n]} \left\{ \frac{G_i \cdot \Omega_i}{\eta_i} \right\} \right)$-approximate Nash Equilibrium, where $\Omega_i = \sup_{x,y \in \Delta(A_i)} \|x - y\|$.*

*Proof.* By Lemma B.3, we have:

$$\mathrm{mReg}_i^T \leq \frac{m_i \bar{R}_i}{\eta_i^T} + m_i \eta_i^1 \cdot c_*^2 \sum_{t=1}^{T} \|v_i^t - v_i^{t-1}\|_\infty^2 - \frac{m_i}{8\eta_i^1} \cdot c^2 \cdot \sum_{t=1}^{T} \left( \|g_i^t - x_i^t\|_1^2 + \|x_i^t - g_i^{t-1}\|_1^2 \right)$$

$$- \frac{m_i}{8\eta_i^1} \sum_{t=1}^{T} \left( \|x_i^t - g_i^t\|^2 + \|x_i^t - g_i^{t-1}\|^2 \right)$$

$$\leq \frac{m_i \bar{R}_i}{\eta_i^T} + m_i \eta_i^1 \cdot c_*^2 (n-1) \sum_{t=1}^{T} \sum_{j \neq i} \|x_j^t - x_j^{t-1}\|_1^2 - \frac{m_i}{16\eta_i^1} \cdot c^2 \sum_{t=1}^{T} \|x_i^t - x_i^{t-1}\|_1^2 - \frac{m_i}{8\eta_i^1} \sum_{t=1}^{T} \left( \|x_i^t - g_i^t\|^2 + \|x_i^t - g_i^{t-1}\|^2 \right)$$

where the second inequality follows from Lemma A.4 and the fact that:

$$\sum_{t=1}^{T} \|x_i^t - x_i^{t-1}\|_1^2 \leq 2\sum_{t=1}^{T} \|x_i^{t-1} - g_i^{t-1}\|_1^2 + 2\sum_{t=1}^{T} \|x_i^t - g_i^{t-1}\|_1^2$$

$$\leq 2\sum_{t=1}^{T} \|x_i^t - g_i^t\|_1^2 + 2\sum_{t=1}^{T} \|x_i^t - g_i^{t-1}\|_1^2,$$

Summing it from 1 to n, we have

$$\sum_{i=1}^{n} \mathrm{mReg}_i^T \leq \sum_{i=1}^{n} \frac{\bar{R}_i m_i}{\eta_i^T} + \left( \eta^1 \bar{m} c_*^2 (n-1)^2 - \frac{\underline{m}c^2}{16\eta^1} \right) \sum_{i=1}^{n} \sum_{t=1}^{T} \|x_i^t - x_i^{t-1}\|_1^2 - \frac{\underline{m}}{8\eta^1} \sum_{i=1}^{n} \sum_{t=1}^{T} \left( \|x_i^t - g_i^t\|^2 + \|x_i^t - g_i^{t-1}\|^2 \right),$$

Since $\eta^1 \leq \frac{c}{4c_*(n-1)}\sqrt{\frac{m}{\overline{m}}}$ and $\sum_{i=1}^{n} \mathrm{mReg}_i^T \geq 0$, we have

$$\sum_{i=1}^{n}\sum_{t=1}^{T}\left(\|x_i^t - g_i^t\|^2 + \|x_i^t - g_i^{t-1}\|^2\right) \leq \sum_{i=1}^{n}\frac{8\bar{R}_i m_i \eta^1}{\eta_i \underline{m}}.$$

Suppose for all $t \in [T]$ that $\sum_{i=1}^{n}\left(\|x_i^t - g_i^t\|^2 + \|x_i^t - g_i^{t-1}\|^2\right) > \epsilon^2$, then

$$\epsilon^2 T \leq \sum_{i=1}^{n}\frac{8\bar{R}_i m_i \eta^1}{\eta_i \underline{m}}.$$

Thus for $T > \frac{1}{\epsilon^2}\sum_{i=1}^{n}\frac{8\bar{R}_i m_i \eta^1}{\eta_i \underline{m}}$, there exists some $t \in [T]$ such that

$$\sum_{i=1}^{n}\left(\|x_i^t - g_i^t\|^2 + \|x_i^t - g_i^{t-1}\|^2\right) \leq \epsilon^2.$$

Thus we get $\|x_i^t - g_i^t\| \leq \epsilon$ and $\|g_i^t - g_i^{t-1}\|^2 \leq 2\|x_i^t - g_i^t\|^2 + 2\|x_i^t - g_i^{t-1}\|^2 \leq 2\epsilon^2$. Observe that the maximization problem associated with (OMD) can be expressed in the following variational inequality form:

$$\langle \eta_i^t \cdot v_i^t - \nabla R_i\left(g_i^t\right) + \nabla R_i\left(g_i^{t-1}\right), z_i - g_i^t \rangle \leq 0, \forall z_i \in \Delta\left(A_i\right), i \in [n].$$

Thus, it follows that

$$\langle v_i^t, z_i - g_i^t \rangle \leq \frac{1}{\eta_i^t} \cdot \|\nabla R_i\left(g_i^t\right) - \nabla R_i\left(g_i^{t-1}\right)\|_* \cdot \|z_i - g_i^t\|$$

$$\leq 2\epsilon\frac{G_i \cdot \Omega_i}{\eta_i},$$

where the first inequality is by the Cauchy-Schwarz inequality, and the last inequality is from $R_i$ is $G_i$ smooth. Moreover, we also have that:

$$|\langle v_i^t, x_i^t - g_i^t \rangle| \leq \|v_i^t\|_* \cdot \|x_i^t - g_i^t\| \leq \epsilon \cdot c_*,$$

Where we use the fact that $\|x_i^t - g_i^t\| \leq \epsilon$, and that $\|v_i^t\|_\infty \leq 1$ (by the normalization hypothesis), next we have that:

$$\langle v_i^t, x_i^t \rangle \geq \langle v_i^t, g_i^t \rangle - \epsilon \cdot c_*$$

$$\geq \langle v_i^t, z_i \rangle - \epsilon \cdot \left(c_* + \frac{2G_i \cdot \Omega_i}{\eta_i}\right),$$

for any $z_i \in \Delta\left(A_i\right)$ and player $i \in [n]$. So the proof follows by definition of approximate Nash equilibria. $\qquad\square$

**Theorem 6.2.** *Suppose $x^t$ is the sequence of strategies output by OMD and the conditions specified in Theorem 6.1 are satisfied, then $x^t$ converges to the set of Nash equilibrium of the game.*

*Proof.* Suppose not, then there exists an open set $U$ contains the set of Nash equilibrium of the game, and a subsequence $x^{t_k} \notin U$. From the proof of Theorem 6.1, with $\eta^1 \leq \frac{c}{4c_*(n-1)}\frac{m}{\overline{m}}$ and $\sum_{i=1}^{n} \mathrm{mReg}_i^T \geq 0$, we have

$$\sum_{i=1}^{n}\sum_{t=1}^{T}\left(\|x_i^t - g_i^t\|^2 + \|x_i^t - g_i^{t-1}\|^2\right) \leq \sum_{i=1}^{n}\frac{8\bar{R}_i m_i \eta^1}{\eta_i \underline{m}}.$$

So for any $\epsilon > 0$, there exists some $T(\epsilon) > 0$ such that $\|x_i^t - g_i^t\| \leq \epsilon$ and $\|x_i^t - g_i^{t-1}\| \leq \epsilon$ for all $t \geq T, i \in [n]$.

This implies that any $x^t$ with $t \geq T(\epsilon)$ will be an $O(\epsilon)$-approximate Nash equilibrium. Further, as $\Delta(A)$ is compact, $x^{t_k} \in \Delta(A)$ is bouned, so without loss of generality, after pass a subsequence, we can assume that $x^{t_k}$ convergences to some $x_\infty \in \Delta(A)$.

Since $\forall \epsilon > 0$, there exists some $k > 0$ such that $\forall \ell \geq k$, $x^{t_\ell}$ is an $\epsilon$-approximate Nash equilibrium, we have that $\forall z \in \Delta(A)$, $\langle v(x^{t_\ell}), z - x^{t_\ell} \rangle \leq \epsilon$.

Taking $l$ to infinity, we have

$$\langle v(x_\infty), z - x_\infty \rangle \leq \epsilon, \quad \forall z \in \Delta(A), \forall \epsilon > 0.$$

Therefore, we have

$$\langle v(x_\infty), z - x_\infty \rangle \leq 0, \quad \forall z \in \Delta(A).$$

Hence $x_\infty$ must be a Nash equilibrium, and $x^{t_k}$ convergences to $x_\infty$, contradict to $x^{t_k} \notin U$. $\qquad \square$

**Lemma B.5.** *Suppose $\{x^t\}_{t=1}^\infty \in \Delta(A)$ converges to a finite set of $E = \{y_1, \ldots, y_\iota\}$ and $\forall \epsilon > 0$, there exists $T \in \mathbb{N}^+$ such that for all $t \geq T$, $\|x^{t+1} - x^t\| \leq \epsilon$, then $\{x^t\}_{t=1}^\infty$ converges to a $y_j \in E$.*

*Proof.* If $E$ only has one point,Then the statement is trivial.So,we suppose $E$ at least has two points. Let

$$d = \inf_{j \neq k} \|y_j - y_k\| > 0, \quad E_k = \left\{ y \in \Delta(A) \mid \|y - y_k\| < \frac{d}{3} \right\}.$$

Since $\{x^t\}_{t=1}^\infty$ converge to $E$, and $\cup_{k=1}^\iota E_k$ is an open set that contains $E$, there are at most finite $x^t$ that are not in $\cup_{k=1}^\iota E_k$. Without lose of generality, after filtering out these finite $x_t$, we can assume $x^t \in \cup_{k=1}^\iota E_k$ for all $t \geq 1$.

Take $\epsilon = d/4$, then there exists $T \in \mathbb{N}^+$ such that $\forall t \geq T$, $\|x^{t+1} - x^t\| < d/4$. Without lose of generality, assume $x^T \in E_j$, then if $x^{T+1} \not\in E_j$, asuume $x^{T+1} \in E_k$, we have

$$\|x^{T+1} - x^T\| \geq \|y_j - y_k\| - \|y_j - x^T\| - \|y_k - x^{T+1}\|$$
$$\geq \frac{d}{3}.$$

As $\|x^{T+1} - x^T\| < \frac{d}{4}$, we must have $x^{T+1} \in E_j$ and similarly $x^t \in E_j$, for all $t \geq T$.

For every convergent subsequence that $x^{t_k} \to x_\infty$ as $k \to \infty$, we have $x_\infty \in \overline{E_j}$. Since $x^t$ converges to $E$, $x_\infty \in E \cap \overline{E_j}$, we have $x_\infty = y_j$ and therefore $x^t$ converges to $y_j$ $\qquad \square$

**Lemma B.6.** *Let $E$ be the set of Nash equilibrium of a norm-form game, then the following are equivalent*

  1. *$E$ is a finite set.*

  2. *$d = \inf_{x,y \in E, x \neq y} \|x - y\| > 0$.*

*Proof.* The proof for the first statement to the second statement is trivial, so we focus on the other direction in this proof.

Suppose $E$ is not a finite set. Then as $E \subseteq \Delta(A)$ is bounded, there exists a cluster point $x$ of $E$ and we next show that $x \in E$ is a Nash equilibrium.

As $x$ is a cluster point of $E$, there exists $\{x^t\}_{t=1}^\infty \in E$ with $x^t \neq x$ such that $x^t \to x$. Because $x^t$ is a Nash equilibrium, we have

$$\langle v(x^t), z - x^t \rangle \leq 0, \forall z \in \Delta(A).$$

By having $t \to \infty$, we have

$$\langle v(x), z - x \rangle \leq 0, \forall z \in \Delta(A).$$

This thus implies that $x \in E$ and

$$d = \inf_{x,y \in E, x \neq y} \|x - y\| \leq \|x - x^t\| \to 0,$$

as $t \to \infty$. Then $d = 0$ and this completes the proof. $\qquad \square$

**Theorem 6.3.** *Suppose $x^t$ is the sequence of strategies generated by OMD and the conditions specified in Theorem 6.1 are satisfied. If the set of Nash equilibrium of the game is discrete, then $x^t$ converges to a Nash equilibrium of the game.*

*Proof.* By Lemma B.6, $d = \inf_{x,y \in E, x \neq y} \|x - y\| > 0$ implies that $E$ is a finite set.

Using the same logic as the proof of Theorem 6.1, for any $\epsilon > 0$, we can find $T(\epsilon) > 0$ such that $\forall t \geq T$,

$$\left\| x^t - g^t \right\|^2 + \left\| x^t - g^{t-1} \right\|^2 \leq \epsilon^2 \, .$$

So

$$\left\| x^t - x^{t-1} \right\|^2 \leq \left\| x^t - g^{t-1} \right\|^2 + \left\| x^{t-1} - g^{t-1} \right\|^2$$
$$\leq 2\epsilon^2 \, .$$

For $\forall t \geq T + 1$, with Theorem B.1, $\{x^t\}_{t=1}^{\infty}$ satisfy the conditions for Lemma B.5. Therefore, $x^t$ converges to a Nash equilibrium of the game. $\qquad \square$

## C   ANALYSIS FOR OFTRL

**Theorem 6.4.** *If each player employs OFTRL with*

- *a pair of norms such that $\|x\| \geq c\|x\|_1$, $\|x\|_* \leq c_*\|x\|_\infty$ for some constant $c, c_*$ and for any $x$,*

- *a $G_i$ smooth regularizer $R_i$, and $R_i$ is Legendre with domain $D_i \subseteq \Delta(A_i)$*

- *learning rate $\eta \leq \frac{c}{4c_*(n-1)}\sqrt{\frac{\underline{m}}{\bar{m}}}$ , where $\eta = \max_i \eta_i$ , $\underline{m} = \min m_i$ , $\bar{m} = \max m_i$.*

*Then if the game has non-negative weight regret, for any $\epsilon > 0$, after $T > \frac{1}{\epsilon^2}\sum_{i=1}^{n}\frac{8\bar{R}_i m_i \eta}{\eta_i \underline{m}}$ iterations, there exists an iterate $\hat{x}^t$ that is an $\epsilon \cdot \left(c_* + 2\max_i\left\{\frac{G_i \cdot \Omega_i}{\eta_i}\right\}\right)$-approximate Nash Equilibrium, where $\Omega_i = \sup_{x,y\in\Delta(A_i)}\|x-y\|$ .*

*Proof.* Let $x^t$ denotes the OMD's sequence iterate that are produced with the same learning rate and $R_i$ to that considered of the OFTRL's. We first show that $\hat{x}^t = x^t$ for all of $t \geq 1$ by induction.

When $t = 1$, $\hat{x}_i^1$ given by equation 2, we have $\nabla R_i(\hat{x}_i^1) = \eta_i \hat{v}_i^0 = \eta_i v_i^0$. For $x_i^1$ given by the OMD update rule, we have

$$\nabla R_i(x_i^1) = \eta_i v_i^0 + \nabla R_i(g_i^0) = \eta_i v_i^0 .$$

As the gradient of the Legendre function is invertible, we have $\hat{x}_i^1 = x_i^1$ for any $i \in [n]$.

Suppose that we have $\hat{x}^s = x^s$, for all $s \leq t$. we next show that $\hat{x}^{t+1} = x^{t+1}$. By the update rule of OFTRL and OMD, we have

$$\nabla R_i(\hat{x}_i^{t+1}) = \eta_i \left(\hat{v}_i^t + \sum_{s=1}^{t}\hat{v}_i^s\right)$$

$$= \eta_i \left(v_i^t + \sum_{s=1}^{t}v_i^s\right)$$

$$\nabla R_i(x_i^{t+1}) = \eta_i v_i^t + \nabla R_i(g_i^t)$$

$$= \eta_i v_i^t + \eta_i v_i^t + \nabla R_i(g_i^{t-1})$$

$$= \eta_i \left(v_i^t + \sum_{s=1}^{t}v_i^s\right) .$$

Therefore, we have $\hat{x}^{t+1} = x^{t+1}$ and we have proved the claim through induction.

With this, the rest of the Theorem follows by using Theorem 6.1. $\square$

**Theorem 6.5.** *Suppose $\hat{x}^t$ is the sequence of strategies output by OFTRL and the conditions specified in Theorem 6.1 are satisfied. If the set of Nash equilibrium of the game is discrete, then $\hat{x}^t$ converges to a Nash equilibrium of the game.*

*Proof.* Since $\hat{x}^t = x^t, \forall t \geq 1$ by the proof of Theorem 7.3, and by Theorem 7.2, we can conclude that $\hat{x}^t$ converges to a Nash equilibrium of the game. $\square$

## D   LEARNING WITH CORRUPTIONS

**Lemma D.1.**

$$\sum_{t=1}^{T}\langle x_i^* - \tilde{x}_i^t, v_i^t\rangle \leq \frac{\bar{R}_i}{\eta_i^T} + \sum_{t=1}^{T}\eta_i^t\|v_i^t - v_i^{t-1}\|_*^2 - \frac{1}{4}\sum_{t=1}^{T}\frac{1}{\eta_i^t}\left(\|g_i^t - \tilde{x}_i^t\|^2 + \|\tilde{x}_i^t - g_i^{t-1}\|^2\right), \forall x_i^* \in \Delta(A_i), i \in [n].$$

*Proof.* The proof is similar to Lemma A.2, so we omit it. $\square$

**Lemma D.2.**

$$\text{mReg}_i^T \leq \frac{m_i \cdot \bar{R}_i}{\eta_i^T} + \sum_{t=1}^{T} m_i \cdot \eta_i^t \|v_i^t - v_i^{t-1}\|_*^2 - \frac{m_i}{4} \sum_{t=1}^{T} \frac{1}{\eta_i^t} \left( \|g_i^t - \tilde{x}_i^t\|^2 + \|\tilde{x}_i^t - g_i^{t-1}\|^2 \right) + \sum_{t=1}^{T} m_i \cdot \|c_i^t\|_1 .$$

*Proof.*

$$\text{mReg}_i^T = \max_{x^* \in \Delta(A_i)} \sum_{t=1}^{T} m_i \langle x^* - \tilde{x}_i^t, v_i^t \rangle + \sum_{t=1}^{T} m_i \langle \tilde{x}_i^t - x_i^t, v_i^t \rangle$$

$$\leq \frac{m_i \cdot \bar{R}_i}{\eta_i^T} + \sum_{t=1}^{T} m_i \cdot \eta_i^t \|v_i^t - v_i^{t-1}\|_*^2 - \frac{m_i}{4} \sum_{t=1}^{T} \frac{1}{\eta_i^t} \left( \|g_i^t - \tilde{x}_i^t\|^2 + \|\tilde{x}_i^t - g_i^{t-1}\|^2 \right) + \sum_{t=1}^{T} m_i \cdot \|c_i^t\|_1 .$$

The last inequality is by Lemma C.1,and since $\|v_i^t\|_\infty \leq 1$. $\qquad\square$

**Theorem 7.1.** *If each player employs OMD under corruption with*

- *a pair of norms such that $\|x\| \geq c\|x\|_1$, $\|x\|_* \leq c_*\|x\|_\infty$ for some constant $c, c_*$ and for any $x$,*

- *$G_i$-smooth regularizer $R_i$,*

- *non-increasing learning rate with $\eta^1 \leq \frac{c}{4(n-1)c_*} \cdot \sqrt{\frac{\underline{m}}{3\bar{m}}}$ , where $\eta^1 = \max_i \eta_i^1$ , $\underline{m} = \min m_i$ , $\bar{m} = \max m_i$ and $\eta_i^t \geq \eta_i > 0$.*

*Then if the game has non-negative weight regret, for any $\epsilon > 0$, after $T > \frac{1}{\epsilon^2} \left\{ \sum_{i=1}^{n} \frac{8m_i \cdot \bar{R}_i \eta^1}{\eta_i \cdot \underline{m}} 48(\eta^1)^2 \cdot \frac{\bar{m}}{\underline{m}} c_*^2 (n-1)^2 \sum_{i=1}^{n} M_i \cdot C_i 8\eta^1 \cdot \frac{\bar{m}}{\underline{m}} \sum_{i=1}^{n} C_i \right\}$ iterations, there exists an iterate $x^t$ that is an $\max_{i \in [n]} \epsilon \cdot \left( c_* + 2 \left\{ \frac{G_i \cdot \Omega_i}{\eta_i} \right\} \right) + \|c_i^t\|_1$-approximate Nash Equilibrium, where $\Omega_i = \sup_{x,y \in \Delta(A_i)} \|x - y\|$ and $M_i = \sup_{t \geq 1} c_i^t$.*

*Proof.* By Lemma D.2, we have

$$\text{mReg}_i^T \leq \frac{m_i \cdot \bar{R}_i}{\eta_i^T} + m_i \cdot \eta_i^1 \cdot c_*^2 \sum_{t=1}^{T} \|v_i^t - v_i^{t-1}\|_\infty^2 - \frac{m_i}{8\eta_i^1} \cdot c^2 \cdot \sum_{t=1}^{T} \left( \|g_i^t - \tilde{x}_i^t\|_1^2 + \|\tilde{x}_i^t - g_i^{t-1}\|_1^2 \right)$$

$$- \frac{m_i}{8\eta_i^1} \sum_{t=1}^{T} \left( \|\tilde{x}_i^t - g_i^t\|^2 + \|\tilde{x}_i^t - g_i^{t-1}\|^2 \right) + \sum_{t=1}^{T} m_i \cdot \|c_i^t\|_1$$

$$\leq \frac{m_i \cdot \bar{R}_i}{\eta_i^T} + m_i \cdot \eta_i^1 \cdot c_*^2 (n-1) \sum_{t=1}^{T} \sum_{j \neq i} \|x_j^t - x_j^{t-1}\|_1^2 - \frac{m_i}{16\eta_i^1} \cdot c^2 \sum_{t=1}^{T} \|\tilde{x}_i^t$$

$$\tilde{x}_i^{t-1}\|_1^2 - \frac{m_i}{8\eta_i^1} \sum_{t=1}^{T} \left( \|\tilde{x}_i^t - g_i^t\|^2 + \|\tilde{x}_i^t - g_i^{t-1}\|^2 \right) + \sum_{t=1}^{T} m_i \cdot \|c_i^t\|_1 ,$$

Where the second inequality is by Lemma A.4 and we use the fact that:

$$\sum_{t=1}^{T} \|\tilde{x}_i^t - \tilde{x}_i^{t-1}\|_1^2 \leq 2 \sum_{t=1}^{T} \|g_i^t - \tilde{x}_i^t\|_1^2 + 2 \sum_{t=1}^{T} \|\tilde{x}_i^t - g_i^{t-1}\|_1^2$$

Summing it from 1 to n :

$$\sum_{i=1}^{n} \mathrm{mReg}_i^T \leq \sum_{i=1}^{n} \frac{m_i \cdot \bar{R}_i}{\eta_i^T} + \bar{m} \cdot \eta^1 \cdot c_*^2 (n-1)^2 \sum_{i=1}^{n} \sum_{t=1}^{T} \|x_i^t - x_i^{t-1}\|_1^2 - \frac{c^2 \cdot m}{16\eta^1} \sum_{i=1}^{n} \sum_{t=1}^{T} \|\tilde{x}_i^t - \tilde{x}_i^{t-1}\|_1^2$$

$$- \frac{m}{8\eta^1} \sum_{i=1}^{n} \sum_{t=1}^{T} \left( \|\tilde{x}_i^t - g_i^t\|^2 + \|\tilde{x}_i^t - g_i^{t-1}\|^2 \right) + \sum_{i=1}^{n} \sum_{t=1}^{T} m_i \cdot \|c_i^t\|_1$$

$$\leq \sum_{i=1}^{n} \frac{m_i \cdot \bar{R}_i}{\eta_i^T} + \left( 3\eta^1 \bar{m} c_*^2 (n-1)^2 - \frac{mc^2}{16\eta^1} \right) \sum_{i=1}^{n} \sum_{t=1}^{T} \|\tilde{x}_i^t - \tilde{x}_i^{t-1}\|_1^2 + 3\bar{m}\eta^1 c_*^2 (n-1)^2 \sum_{i=1}^{n} \sum_{t=1}^{T} \left( \|c_i^t\|_1^2 + \|c_i^{t-1} \right.$$

$$- \frac{m}{8\eta^1} \sum_{i=1}^{n} \sum_{t=1}^{T} \left( \|\tilde{x}_i^t - g_i^t\|^2 + \|\tilde{x}_i^t - g_i^{t-1}\|^2 \right) + \sum_{i=1}^{n} \sum_{t=1}^{T} m_i \cdot \|c_i^t\|_1 \,.$$

Where the last inequality is by the fact that:

$$\|x_i^t - x_i^{t-1}\|_1^2 \leq 3\|x_i^t - \tilde{x}_i^t\|_1^2 + 3\|x_i^{t-1} - \tilde{x}_i^{t-1}\|_1^2 + 3\|\tilde{x}_i^t - \tilde{x}_i^{t-1}\|_1^2$$

Since we choose $\eta^1$ such that $\eta^1 \leq \frac{c}{4(n-1)c_*} \cdot \sqrt{\frac{m}{3\bar{m}}}$ and $\sum_{i=1}^{n} \mathrm{mReg}_i^T \geq 0$ we have

$$\sum_{i=1}^{n} \sum_{t=1}^{T} \left( \|\tilde{x}_i^t - g_i^t\|^2 + \|\tilde{x}_i^t - g_i^{t-1}\|^2 \right) \leq \sum_{i=1}^{n} \frac{8m_i \cdot \bar{R}_i \eta^1}{\eta_i \cdot \underline{m}} + 48(\eta^1)^2 \cdot \frac{\bar{m}}{\underline{m}} c_*^2 (n-1)^2 \sum_{i=1}^{n} M_i \cdot C_i + 8\eta^1 \cdot \frac{\bar{m}}{\underline{m}} \sum_{i=1}^{n} C_i \,.$$

We use $\sum_{t=1}^{T} \|c_i^t\|_1^2 \leq M_i \cdot \sum_{t=1}^{T} \|c_i^t\|_1 \leq M_i \cdot C_i$.

Suppose for all $t \in [T]$ that $\sum_{i=1}^{n} \left( \|\tilde{x}_i^t - g_i^t\|^2 + \|\tilde{x}_i^t - g_i^{t-1}\|^2 \right) > \epsilon^2$, then

$$\epsilon^2 T \leq \sum_{i=1}^{n} \frac{8m_i \cdot \bar{R}_i \eta^1}{\eta_i \cdot \underline{m}} + 48(\eta^1)^2 \cdot \frac{\bar{m}}{\underline{m}} c_*^2 (n-1)^2 \sum_{i=1}^{n} M_i \cdot C_i + 8\eta^1 \cdot \frac{\bar{m}}{\underline{m}} \sum_{i=1}^{n} C_i \,.$$

Thus for $T > \frac{1}{\epsilon^2} \{ \sum_{i=1}^{n} \frac{8m_i \cdot \bar{R}_i \eta^1}{\eta_i \cdot \underline{m}} + 48(\eta^1)^2 \cdot \frac{\bar{m}}{\underline{m}} c_*^2 (n-1)^2 \sum_{i=1}^{n} M_i \cdot C_i + 8\eta^1 \cdot \frac{\bar{m}}{\underline{m}} \sum_{i=1}^{n} C_i \}$, there exists some $t \in [T]$ such that

$$\sum_{i=1}^{n} \left( \|\tilde{x}_i^t - g_i^t\|^2 + \|\tilde{x}_i^t - g_i^{t-1}\|^2 \right) \leq \epsilon^2 \,.$$

Thus we get $\|\tilde{x}_i^t - g_i^t\| \leq \epsilon$ and $\|g_i^t - g_i^{t-1}\|^2 \leq 2\|\tilde{x}_i^t - g_i^t\|^2 + 2\|\tilde{x}_i^t - g_i^{t-1}\|^2 \leq 2\epsilon^2$. Observe that the maximization problem associated with (OMD) can be expressed in the following variational inequality form:

$$\langle \eta_i^t \cdot v_i^t - \nabla R_i(g_i^t) + \nabla R_i(g_i^{t-1}), z_i - g_i^t \rangle \leq 0, \forall z_i \in \Delta(A_i), i \in [n].$$

Thus,it follows that

$$\langle v_i^t, z_i - g_i^t \rangle \leq \frac{1}{\eta_i^t} \cdot \|\nabla R_i(g_i^t) - \nabla R_i(g_i^{t-1})\|_* \cdot \|z_i - g_i^t\|$$

$$\leq 2\epsilon \frac{G_i \cdot \Omega_i}{\eta_i},$$

where the first inequality is by the Cauchy-Schwarz inequality,and the last inequality is from $R_i$ is $G_i$ smooth.Moreover, we also have that:

$$|\langle v_i^t, \tilde{x}_i^t - g_i^t \rangle| \leq \|v_i^t\|_* \cdot \|\tilde{x}_i^t - g_i^t\| \leq \epsilon \cdot c_*,$$

And

$$|\langle v_i^t, \tilde{x}_i^t - x_i^t \rangle| \leq \|v_i^t\|_\infty \cdot \|c_i^t\|_1 \leq \|c_i^t\|_1.$$

Where we use the fact that $\|\tilde{x}_i^t - g_i^t\| \leq \epsilon$, and that $\|v_i^t\|_\infty \leq 1$ (by the normalization hypothesis), next we have that:

$$
\begin{aligned}
\langle v_i^t, x_i^t \rangle &\geq \langle v_i^t, \tilde{x}_i^t \rangle - \|c_i^t\|_1 \\
&\geq \langle v_i^t, g_i^t \rangle - \epsilon \cdot c_* - \|c_i^t\|_1 \\
&\geq \langle v_i^t, z_i \rangle - \epsilon \cdot \left( c_* + \frac{2 G_i \cdot \Omega_i}{\eta_i} \right) - \|c_i^t\|_1,
\end{aligned}
$$

for any $z_i \in \Delta(A_i)$ and player $i \in [n]$. So the proof follows by definition of approximate Nash equilibria. $\qquad\square$

**Theorem 7.2.** *Suppose $x^t$ is the sequence of strategies played with OMD under corruptions and the conditions specified in Theorem 7.1 are satisfied, then $x^t$ converges to the set of Nash equilibria of the game.*

*Proof.* Since $\sum_{t=1}^{\infty} \|c_i^t\|_1 < \infty, \forall i \in [n]$, and by the proof of Theorem 8.1, $\forall \epsilon > 0$, we can find $T(\epsilon) > 0, \forall t \geq T, x^t$ is an $\epsilon$-approximate Nash equilibrium. Therefore, similar to the proof of Theorem B.1, $x^t$ also converges to the set of Nash equilibrium of the game. $\qquad\square$

**Theorem 7.3.** *Suppose $x^t$ is the sequence of strategies played with OMD under corruptions and the conditions specified in Theorem 7.1 are satisfied. If the set of Nash equilibria of the game is discrete, then $x^t$ converges to a Nash equilibrium of the game.*

*Proof.* We only need to proof $\forall \epsilon > 0, \exists T > 0, \forall t \geq T, \|x^t - x^{t-1}\| \leq \epsilon$.

Since $\sum_{t=1}^{\infty} \|c_i^t\|_1 < \infty, \forall i \in [n]$, we can find $T_i > 0, \forall t \geq T_i - 1, \|c_i^t\| < \frac{\epsilon}{4}$ (The norms of finite dimensional normed linear Spaces are equivalent to each other).

By the proof of Theorem 8.1, there exists a $\hat{T}_i > 0, \forall t \geq \hat{T}_i$,

$$
\sum_{i=1}^{n} \left( \|\tilde{x}_i^t - g_i^t\|^2 + \|\tilde{x}_i^t - g_i^{t-1}\|^2 \right) \leq \frac{\epsilon^2}{8} \cdot
$$

Thus,

$$
\begin{aligned}
\left\| \tilde{x}_i^t - \tilde{x}_i^{t-1} \right\|^2 &\leq \left\| \tilde{x}_i^t - g^{t-1} \right\|^2 + \left\| \tilde{x}_i^{t-1} - g^{t-1} \right\|^2 \\
&\leq \frac{\epsilon^2}{4} \cdot
\end{aligned}
$$

Let $T = \max_{i \in [n]} \{ T_i, \hat{T}_i \}$, we have that for $\forall t \geq T$:

$$
\begin{aligned}
\left\| x_i^t - x_i^{t-1} \right\| &\leq \|c_i^t\| + \left\| \tilde{x}_i^t - \tilde{x}_i^{t-1} \right\| + \|c_i^{t-1}\| \\
&\leq \epsilon \cdot
\end{aligned}
$$

$\qquad\square$

**Theorem 7.4.** *If each player employs OFTRL with corruption with*

- *a pair of norms such that $\|y\| \geq c\|y\|_1$, $\|y\|_* \leq c_* \|y\|_\infty$ for some constant $c, c_*$ and for any $y$,*

- *a $G_i$ smooth regularizer $R_i$, and $R_i$ is Legendre with domain $D_i \subseteq \Delta(A_i)$*

- *learning rate $\eta \leq \frac{c}{4 c_* (n-1)} \sqrt{\frac{\underline{m}}{3\bar{m}}}$, where $\eta = \max_i \eta_i$, $\underline{m} = \min m_i$, $\bar{m} = \max m_i$.*

*Then if the game has non-negative weight regret, for any $\epsilon > 0$, after $T > \frac{1}{\epsilon^2} \left\{ \sum_{i=1}^{n} \frac{8 m_i \cdot \bar{R}_i \eta^1}{\eta_i \cdot \underline{m}} + 48(\eta^1)^2 \cdot \frac{\bar{m}}{\underline{m}} c_*^2 (n-1)^2 \sum_{i=1}^{n} M_i \cdot C_i + 8\eta^1 \cdot \frac{\bar{m}}{\underline{m}} \sum_{i=1}^{n} C_i \right\}$ iterations, there exists an iterate $y^t$ that is an $\max_{i \in [n]} \epsilon \cdot \left( c_* + 2 \left\{ \frac{G_i \cdot \Omega_i}{\eta_i} \right\} \right) + \|c_i^t\|_1$-approximate Nash Equilibrium, where $\Omega_i = \sup_{x,y \in \Delta(A_i)} \|x - y\|$ and $M_i = \sup_{t \geq 1} c_i^t$.*

*Proof.* Define $x_i^0 = g_i^0 = \operatorname{argmin}_{x_i \in \Delta(A_i)} R_i(x_i)$,Let:

$$\tilde{x}_i^{t+1} = \operatorname*{argmax}_{x_i \in \Delta(A_i)} \eta_i \langle x_i, v_i^t \rangle - D_{R_i}(x_i, g_i^t),$$

$$x_i^{t+1} = \tilde{x}_i^{t+1} + c_i^{t+1},$$

$$v_i^{t+1} = v_i(x^{t+1})$$

$$g_i^{t+1} = \operatorname*{argmax}_{g_i \in \Delta(A_i)} \eta_i \langle g_i, v_i^{t+1} \rangle - D_{R_i}(g_i, g_i^t).$$

We first simultaneous show that $\tilde{y}^t = \tilde{x}^t$ and $y^t = x^t$ for all of $t \geq 1$ by induction.

When $t = 1$, we have $\nabla R_i(\tilde{y}_i^1) = \eta_i \hat{v}_i^0 = \eta_i v_i^0$. For $\tilde{x}_i^1$ given by our definition, we have

$$\nabla R_i(\tilde{x}_i^1) = \eta_i v_i^0 + \nabla R_i(g_i^0) = \eta_i v_i^0.$$

As the gradient of the Legendre function is invertible, we have $\tilde{y}_i^1 = \tilde{x}_i^1$ for any $i \in [n]$.Thus $y_i^1 = x_i^1$ for any $i \in [n]$.

Suppose that we have $\tilde{y}^s = \tilde{x}^s$,and $y_i^s = x_i^s$ for all $s \leq t$. we next show that $\tilde{y}^{t+1} = \tilde{x}^{t+1}$,and $y_i^{t+1} = x_i^{t+1}$. By the update rule of Non-honest OFTRL and Non-honest OMD, we have

$$\nabla R_i(\tilde{y}_i^{t+1}) = \eta_i \left( \hat{v}_i^t + \sum_{s=1}^t \hat{v}_i^s \right)$$

$$= \eta_i \left( v_i^t + \sum_{s=1}^t v_i^s \right)$$

$$\nabla R_i(\tilde{x}_i^{t+1}) = \eta_i v_i^t + \nabla R_i(g_i^t)$$

$$= \eta_i v_i^t + \eta_i v_i^t + \nabla R_i(g_i^{t-1})$$

$$= \eta_i \left( v_i^t + \sum_{s=1}^t v_i^s \right).$$

Therefore, we have $\tilde{y}^{t+1} = \tilde{x}^{t+1}$ and $y_i^{t+1} = x_i^{t+1}$. And we have proved the claim through induction.

With this, the rest of the Theorem follows by using Theorem 8.1. □

**Theorem 7.5.** *Suppose $y^t$ is the sequence of strategies player with OFTRL under corruptions and the conditions specified in Theorem 7.4 are satisfied. If the set of Nash equilibrium of the game is discrete, then $y^t$ converges to the a Nash equilibrium of the game.*

*Proof.* Since $y^t = x^t, \forall t \geq 1$ by the proof of Theorem D.3, and by Theorem D.2, we can conclude that $y^t$ converges to a Nash equilibrium of the game. □

