# OpenReview forum: "Pointwise Convergence in Games with Conflicting Interests"
_ICLR.cc/2026/Conference — ICLR 2026 Conference Withdrawn Submission_

### Official Review · Reviewer_3ngM · 2025-10-29

**Soundness:** 2
**Presentation:** 1
**Contribution:** 3
**Rating:** 4
**Confidence:** 4

**Summary:**

The authors consider a new class of games: games with non-negative weighted regret.
They show that this class includes harmonic games and zero-sum games (which was already known).
They then show optimistic variants of classical no-regret learning algorithms (Optimistic MD and Optimistic FTRL) can find an $\epsilon-$Nash equilibirum in $O(1/\epsilon^2)$ iterations.
Beyond this, they prove asymptotic pointwise convergence of those algorithms to the set of Nash equilibrium, even when the set is discrete.
They also show a set of analogous results for learning with corruptions, given the corruption level remains finite.

**Strengths:**

*   They connect harmonic games to a new class of games with non-negative weighted regret. By doing so, they are able to obtain a more comprehensive set of convergence results than previous work, with a simpler analysis.
*   Their pointwise convergence of the algorithms to the set of Nash equilibria is the first result of this kind for harmonic games.

**Weaknesses:**

*   It seems that the pointwise convergence is the major contribution of this paper. I have two concerns about the pointwise convergence results: (1) based on my understanding, the proof of the asymptotic and pointwise convergence is somehow not hard when we have $\lVert x^t - g^t \rVert^2 + \lVert x^t - g^{t-1} \rVert^2 → 0$ as $t → 0$; (2) I am not very sure how interesting the pointwise convergence is - the asymptotic convergence result is mentioned in Remark A.15 of Anagnostides et al. (2022).
*   This paper needs improvement in terms of writing, espacially in the proofs.
*    The authors state "Even in the two-player zero-sum game, the pointwise convergence result often requires the assumption of unique Nash equilibrium." This is actually the *same* assumption you make. Having a discrete set of Nash equilibria in a two-player zero-sum game is only possible exactly when there is a singly unique equilibrium.
*   The authors do not discuss/explain whether their "discrete set of Nash equilibria" assumption for pointwise convergence is useful. As per the point above, it is not useful for two-player zero-sum games. Is it useful in harmonic games?

**Questions:**

*   My understanding is that Theorem 6.1 establishs a best-iterate convergence rate; I think it should be specified in the paper if that is the case
*   In definition 3.3, should the definition of the Harmonic game involve \mu_{i, b_i} instead of \mu_{i, a_i}?
*   In the proof of Theorem 6.3. (p17), should that be $\lVert x^t - x^{t-1} \rVert^2 \leq 2 \lVert x^t - g^{t-1} \rVert^2 + 2 \lVert x^{t-1} - g^{t-1} \rVert^2$?
*   What is the harmonic game in the experiment section?
* I think asymptotic pointwise convergence is already implied by the results in Anagnostides et al. 2022 for zero-sum games? You can go from a duality gap bound to a distance bound through metric subregularity. If so then the authors need to change Table 1 amongst other things.

Minor quibbles:
*   Consider using brackets to separate the off-text references;
*   In theorem 6.1 and other theorems, it is better to specify that $\eta_i$ is the lower bound of all adaptive stepsizes to avoid confusion;
*   Typos
    *   Definition 3.1: \mathcal{A} → A
    *   Definition 3.3: "Norm-form" → "Normal-form"
    *   p4 l168: Abdou et al. (2022) [show?] that
    *   p12 Eq. (3): $x_{i.a_i} → x_{i,a_i}$
    *   p15 l808-809: "bouned" → "bounded", "pass" → "passing", "convergences" → "converges"
    *   p16 l832: "asuume" → "assume"

---

### Official Review · Reviewer_7UkH · 2025-10-31

**Soundness:** 3
**Presentation:** 1
**Contribution:** 2
**Rating:** 2
**Confidence:** 2

**Summary:**

The paper introduces a new class of games that are characterized by a non-negative weighted regret. The proposed class is a direct generalization of a previously proposed class of games with non-negative regret. The paper proves that the proposed class of games contains harmonic games and some classes of zero-sum games, such as two-player zero-sum games. Furthermore, the paper looks at two existing algorithms, namely the optimistic variants of Mirror Descent (OMD) and Follow the Regularized Leader (OFTRL) and proves that these two algorithms converge in games with non-negative weighted regret at a rate of $1 / \epsilon^2$ and also proves pointwise convergence of the two algorithms. Lastly, the paper proves that the convergence guarantees remain unchanged even if players are allowed to deviate from their strategies by a finite amount.

While the paper’s ideas are conceptually interesting and aim to bridge the gap between harmonic and zero-sum games, the paper suffers from significant clarity issues. The main strengths are the generalization and the extension of convergence guarantees to a broader class of games, which could be of theoretical importance for the study of learning dynamics in games. However, the paper’s clarity and presentation are major weaknesses. Many terms are left undefined, and the paper provides insufficient context and discussion of prior work, making it difficult to assess the novelty and significance of the contributions. The empirical evaluation is minimal and unconvincing. Lastly, the appendix is poorly formatted, with missing lemma statements, proof overflows, and serious readability issues. The paper also has numerous typos and inconsistencies in terminology.

**Strengths:**

**Originality.** The paper extends an existing class of games, previously limited to subclasses of zero-sum games, to include harmonic games, thus introducing a broader and more general theoretical framework.

**Soundness.**
The paper rigorously establishes and proves convergence guarantees for two classical no-regret algorithms.
It demonstrates that these guarantees are maintained even when we allow player deviations, reinforcing the robustness.

**Weaknesses:**

**Originality.**
The paper provides insufficient discussion of prior work, making it unclear what the paper’s contributions are and what is considered prior work. The lack of context obscures the original contribution relative to earlier formulations of similar game classes.

**Clarity.**
Many parts of the paper are difficult to follow due to undefined terminology (e.g., “pointwise convergence”) and imprecise explanations, which require multiple readthroughs to get a solid understanding of the presented concepts.
The appendix is poorly written, with lemmas missing their statements, countless overflows, and proofs of Theorems 6.1 and 7.1 overflowing over the right edge of the document, making it impossible to follow the proofs.

**Soundness.**
The experiment section is limited, containing only two overly simplistic games (one of them being a two-player zero-sum game, which is trivial to solve, and it is well-known that the algorithms converge there).
The paper completely lacks any empirical evaluation of claims and theorems from Section 7, which would strengthen the paper’s empirical evaluation considerably.

**Questions:**

Was the convergence rate of the OFTRL algorithm in games with non-negative regret known before or is that a new result?

Minor comments:
* Countless typos and grammatical errors throughout the paper
* Line 41: There’s a missing third component – the non-strategic subspace
* Definition 4.1: The definition of regret should only consider pure actions, not mixed strategies
The terms non-negative weighted regrets and weighted regrets are used interchangeably, which creates unnecessary confusion (also weighted vs. weight regret)

---

### Official Review · Reviewer_Ewui · 2025-11-01

**Soundness:** 2
**Presentation:** 1
**Contribution:** 2
**Rating:** 2
**Confidence:** 4

**Summary:**

This paper studies a class of games with non-negative *weighted* regrets, which extends the notion of games with non-negative regrets from [Anagnostieds+22] to a weighted regime. The main contributions of the paper are to prove for this class of games O(1/\sqrt{T}) best-iterate convergence to Nash using optimistic FTRL and OMD (using smooth regularizers), and asymptotic last-iterate convergence.

**Strengths:**

The paper attempts to extend the notion of games with nonnegative regrets to a weighted regime.

**Weaknesses:**

Overall, I believe the contribution of the paper is very narrow, and the presentation and writing requires significant improvements.

Regarding the significance of the contributions: the main contribution of the paper is to introduce a notion of games with non-negative weighted regret (which extends the non-negative regret notion from [Anagnostides+22] to a weighted regime) and to study properties of optimistic learning algorithms in these games. Evaluating the significance of results then relies on two components: (a) how significant is the class of games relative to prior known classes and (b) do the convergence guarantees of the algorithms significantly change, and/or are there substantial new technical tools that are needed to obtain such guarantees.

Regarding (a), as the authors mention, the class of games non-negative (unweighted) regret are of course a subset of the class of games with non-negative weighted regret. This immediately implies that all the (best-iterate) convergence results in the present paper (in particular, Theorems 6.1 and 6.4) for games with non-negative weighted regret are already captured in the work of [Anagnostides+22]. On the other hand, the present paper does show that the class of harmonic games has non-negative weighted regrets (Lemma 5.1). Without a concrete description of the actual example, the authors claim that there exists a harmonic game instance with negative (unweighted) regret -- but without a formal description of the game and a proof of such a claim, this is not completely obvious. However, supposing that this informal claim is true, then the results of Theorem 6.1 (and Theorem 6.4) do establish the first non-asymptotic convergence rates for harmonic games. On the other hand, regarding the point (b) above, the technical novelty in obtaining the main results still appears marginal relative to the results of [Anagnostides+22]. In particular, the proof of Theorem 6.1 in Appendix B follows almost identically to that of Theorem 3.4 of [Anagnostides+22] (see the proof in Appendix A of that paper). For these reasons, I believe the significance of the technical contributions to be limited.

Additionally, the presentation and discussion of the main results could be improved for better interpretation. For example:
* The statements of Theorems 6.1 and 6.4 give *best-iterate* convergence rates to Nash. This is in contrast to the asymptotic last-iterate convergence guarantees of Theorems 6.2 and 6.5. I believe the nature of this non-asymptotic convergence is not distinguished enough when describing the paper's contributions in the introduction, and in Table 1. In other words: the presentation would be improved by more clearly distinguishing the nature of convergence guarantees that are obtained. This also applies when discussing the prior results of [Anagnostides+22] (who also establish best-iterate convergence for games with non-negative regret), for example in Section 4, Lines 223-225.
* Theorems 6.1 and 6.4 additionally assume smooth and Legendre regularizers. For Theorem 6.1 (OMD), it is not clear where in the proof the Legendre property is used. Moreover, the smoothness assumption precludes the use of regularizers like negative-entropy (note that this is mentioned also in [Anagnostides+22]). The present paper is lacking a sufficient discussion on why these assumptions are required, which again would enhance the presentation and interpretation of the results.
* Details on the experimental results are quite lacking: for instance, which regularizers are used to implement OMD and OFTRL? Moreover, both OMD and OFTRL are already known to have linear last-iterate convergence on matching pennies (e.g., from [Wei+2020]), and thus this empirical evaluation again does not seem to make a novel contribution.

For some further (low-level) recommended changes and typos:
* L116: "followed the regularized leader" --> "follow the regularized leader"
* L174: "A Norm-form game" --> "A Normal-form game"
* There are some inconsistencies in the capitalization of "follow the regularized leader" and "mirror descent" (e.g., comparing lines 200-202 with, e.g., 77-79) -- I would recommend being consistent with one style.
* The proof of Theorems 6.1 and 7.1 in the appendix should be fixed: currently, there are displayed equations that are rendered off the page (e.g., on pages 14 and 20).
* References [Leonardos, 2022a] and [Leonardos, 2022b] are the same.

**Questions:**

Can you provide the exact harmonic game instance from Figure 1, as well as a proof of its negative regret?

---

### Official Review · Reviewer_UmJ3 · 2025-11-01

**Soundness:** 1
**Presentation:** 3
**Contribution:** 1
**Rating:** 2
**Confidence:** 5

**Summary:**

This paper considers learning in a class of normal form games with "weighted non-negative regret". This is a weighted variant of a class of games considered by Anagnostides et al. (2022), and contains as special cases the class of two-player zero-sum games and harmonic games. The authors consider two optimistic learning algorithms, optimistic mirror descent (OMD) and optimistic FTRL (OFTRL), and they show that, if either algorithm is run with a Lipschitz smooth regularizer (which must also be Legendre for OFTRL) and a small enough step-size, the induced sequence of play converges to the set of Nash equilibria of the game. Moreover, the authors provide a bound on how many iterations will be required to produce an iterate that is an $\varepsilon$-equilibrium, and they show that if the game only admits a discrete set of Nash equilibria, then the convergence result above can be refined to convergence to a point. Finally, they also consider the case where the players may be playing off-algorithm up to a small corruption level, and they show that the above results still hold if the corruption level is small enough.

The authors' presentation is solid, but there are significant issues with the soundness of their results, which in turn severely limit the contributions of the paper. My "reject" recommendation should not be interpreted as a harsh criticism of the paper, but as my personal assessment of the gap that the paper would have to close in order to be publishable at a top-tier venue. I explain all this in detail in the "weaknesses" section below.

**Strengths:**

The paper is treating an interesting and relevant topic. The realization that the property of "non-negative weighted regret" is what lies behind a fair number of results in the literature is an interesting one.

**Weaknesses:**

Unfortunately, the paper has a number of important issues:
1. The authors are sometimes going too fast with their definitions and assumptions—for example, the Bregman divergence is never defined, notions and statements like "$G$-smooth", "Legendre" or "the corruption level remains finite" are not explained, etc. The expert reader can probably navigate through some of these but, overall these omissions can be quite dangerous—see below.
2. In Theorem 6.4 (and, later, in Theorem 7.4), the authors posit that the regularizer $R$ is Lipschitz smooth and Legendre. Unfortunately, these assumptions are mutually incompatible: the former means that the gradient of the regularizer is (Lipschitz) continuous, and hence finite over the entire simplex (as in the case of the Euclidean regularizer $R(x) = \|x\|_2^2/2$); the latter posits that the regularizer becomes infinitely "steep" near the boundary of the simplex, that is, $\|\nabla R(x_n)\| \to \infty$ whenever $x_n$ converges to the boundary of the simplex (e.g., as in the case of the entropic regularizer, $R(x) = \sum_j x_j \log x_j$). As such, the entire analysis for OFTRL is vacuous. [This is an is an issue where the lack ]
3. While on this, it should be noted that the Lipschitz smoothness assumption excludes widely used cases like optimistic multiplicative weights and the like. I would have expected the authors to discuss this (as well as the relation between OMD and OFTRL) but, unfortunately, they didn't (an omission which is partially responsible for the previous point).
4. During my first read of the paper, I was very much surprised by Theorem 7.3, which roughly states that if "the conditions specified in Theorem 7.1 are satisfied" (corruption level upper bounded by $M$ and step-size suffiicently small), then the OMD sequence of play converges to the game's set of Nash equilibria. If true, this would be very surprising, because OMD could output a Nash equilibrium at each instance of play (since this is a fixed point of the algorithm), but due to corruptions, the players might always be a bounded distance away from it. I was struggling to make sense of this, until I saw in the proof of Theorem 7.2 in the appendix (L1095) that the authors are making an extra assumption, namely that the sequence of corruptions is summable—that is, the corruption level is vanishing over time in a summable manner. This is a drastically different assumption than "the conditions specified in Theorem 7.1", and it makes the theorem much easier to understand and obtain.
5. I also noted that much of the authors' analysis is much closer to previous work than suggested in the paper: almost all of the calculations in Lemma 5.1 (except the very last string of inequalities) is present in Legacci et al., Lemmas B.1—B.3 are also reshufflings of statements present in Legacci et al., Lemma B.4 is taken directly from Anagnostides et al. (2022) (the authors cite this), etc. Moreover, the "best iterate" technique of leveraging a near-equilibrium point from a low-regret guarantee is well-known in the literature, so Theorem 6.1 is not obtained using "drastically different" techniques relative to the existing literature.
6. Finally, the authors state as a major contribution Theorem 6.5 which states that "if the set of Nash equilibria is discrete, the algorithm converges to a point". However, the assumption of isolated Nash equilibria essentially trivializes the matter: if the limit points of the algorithm are all Nash equilibria (which is the previous theorem), and Nash equilibria are isolated, it is trivial to see that the algorithm cannot jump from one to another, so the claim follows trivially.

**Questions:**

I am stating below a few questions:
1. Can the authors provide an application-relevant example of a game with non-negative weighted regret which is not a two-player zero-sum game, a harmonic game, or strategically equivalent to one? [Anagnostides et al. (2022) provides examples of polymatrix games, but these are not mixed extensions of normal form games]
2. Can the authors provide an example of a game with non-negative weighted regret and a set of isolated Nash equilibria? Providing such an example would be crucial in ensuring that the discreteness assumption of Theorem 6.5 is not mutually exclusive with non-negative weighted regret. [In zero-sum games for example, the set of equilibria is always convex, so it cannot be isolated]
3. I believe that what really matters is not the property of "non-negative weighted regret", but the fact that there is a strategy profile $x^\ast$ and non-negative weights $m_i$ such that $\sum_{i} m_i \langle x_i^\ast - x_i, v_i(x) \rangle \geq 0$ for all $x$, i.e., that the game satisfies a "weighted" Minty-type condition. Can the authors comment on this? [In particular, are you aware of an example of a game that satisfies the "non-negative weighted regret" property without satsifying the above weighted Minty condition?]

---

### Note · Authors · 2025-11-18

**Comment:**

We thank the reviewer for the constructive comments, and we will work on improving the work for a future venue.

**Withdrawal Confirmation:**

I have read and agree with the venue's withdrawal policy on behalf of myself and my co-authors.